**Implementation of CCDC to produce the LCMAP Collection 1.0 annual land surface change product**

George Z. Xian[1], Kelcy Smith[2], Danika Wellington[2], Josephine Horton[2], Qiang Zhou[3], Congcong Li[3], Roger Auch[1], Jesslyn F. Brown[1], Zhe Zhu[4], and Ryan R. Reker[2]

[1]United States Geological Survey (USGS) Earth Resources Observation and Science (EROS) Center, Sioux Falls, South Dakota 57198, U.S.A.

[2]KBR, Contractor to the USGS EROS Center, Sioux Falls, SD, 57198, U.S.A.

[3]ASRC Federal Data Solutions (AFDS), Contractor to the USGS EROS Sioux Falls, SD 57198, U.S.A.

[4]Department of Natural Resources and the Environment, University of Connecticut, Storrs, CT, U.S.A.

Correspondence: George Xian (xian@usgs.gov)

**Abstract**

The increasing availability of high-quality remote sensing data and advanced technologies has spurred land cover mapping to characterize land change from local to global scales. However, most land change datasets either span multiple decades at a local scale or cover limited time over a larger geographic extent. Here, we present a new land cover and land surface change dataset created by the Land Change Monitoring, Assessment, and Projection (LCMAP) program over the conterminous United States (CONUS). The LCMAP land cover change dataset consists of annual land cover and land cover change products over the period 1985-2017 at 30-m resolution using Landsat and other ancillary data via the Continuous Change Detection and Classification (CCDC) algorithm. In this paper, we describe our novel approach to implement the CCDC algorithm to produce the LCMAP product suite composed of five land cover and five land surface change related products. The LCMAP land cover products were validated using a collection of ~ 25,000 reference samples collected independently across CONUS. The overall agreement for all years of the LCMAP primary land cover product reached 82.5%. The LCMAP products are produced through the LCMAP Information Warehouse and Data Store (IW+DS) and Shared Mesos Cluster systems that can process, store, and deliver all datasets for public access. To our knowledge, this is the first set of published 30-m annual land change datasets that include land cover, land cover change, and spectral change spanning from the 1980s to the present for the United States. The LCMAP product suite provides useful information for land resource management and facilitates studies to improve the understanding of terrestrial ecosystems and the complex dynamics of the Earth system. The LCMAP system could be implemented to produce global land change products in the future.

## 1 Introduction

Changes in land cover and land surface are one of the greatest and most immediate influences on the Earth system, and these changes will continue in association with a surging human population and growing demand on land resources (Szantoi et al., 2020). Changes in land cover and ecosystems and their implications for global environmental change and sustainability are major research challenges for developing strategies to respond to ongoing global change while meeting development goals (Turner II et al., 2007). Unknowns related to the spatial extent and degrees of impacts of anthropogenic activities on natural systems and strategies to respond to ongoing global change hinder efforts to overcome sustainability challenges (Erb et al., 2017; Reid et al., 2010). An improved understanding of the complex and dynamic interactions between the various Earth system components, including humans and their activities, is critical for policymakers and scientists (Foley, 2005; Foley et al., 2011). To fully understand these processes and monitor these changes, accurate and frequently updated land cover information is essential for scientific research and to assist decision makers in responding to the challenges associated with competing land demands and land surface change.

The characteristics of land surface fundamentally connect with the functioning of Earth's terrestrial surface. Satellite observations have been used to observe the Earth's surface and to characterize land cover and change from local to global scales. Remote sensing data allows us to obtain information over large areas in a practical and accurate manner. With advanced technologies and accumulating satellite data, countries and regions have produced multi-spatial and multi-temporal resolution land cover products (Chen et al., 2015; Gong et al., 2020; Hansen, 2013; Homer et al., 2020; Li et al., 2020). A variety of land change mapping has been carried out to produce land cover and change products in the United States. Among these efforts are the widely known National Land Cover Database (NLCD) products. NLCD has provided comprehensive, general-purpose land cover mapping products at 30-m resolution since 2001 in the United States, and the products have been published and updated across more than a decade (Homer et al., 2020). NLCD provides Anderson Level II land cover classification (Anderson, 1976) for the conterminous United States (CONUS) at approximately 2–3-year intervals. Other national-scale mapping projects focus on specific land cover themes. Among these are the Landscape Fire and Resource Management Planning Tools (LANDFIRE)  (Picotte et al., 2019),

which maps vegetation and fuels in support of wildfire management, and the Cropland Data
Layer (Boryan et al., 2011) generated by the National Agricultural Statistics Service (NASS) of
the United States Department of Agriculture (USDA). Due to the need to incorporate data from
neighboring years, as well as extensive post-processing, ancillary dataset dependencies, and
analyst-supported refinement, release dates for both LANDFIRE and NLCD products are
typically several years subsequent to the nominal map year. Other products including national
urban extent change and vegetation phenology data are available (Li et al., 2019; Li et al., 2020).
These projects vary in how land change information is incorporated or expressed across product
releases. Continuous data stacks allow for an increase in input features for land cover
classification. Frequent data also provides the opportunity for near-real time change monitoring
with frequently updated image acquisitions. The availability of land change information has led
to approaches that attempt to monitor surface properties continuously through time (Franklin et
al., 2015; Gong et al., 2019; Hermosilla et al., 2018; Homer et al., 2020; Kennedy et al., 2015; Li
et al., 2020). Such approaches have several advantages over traditional image processing
techniques based on small numbers of images (Bullock et al., 2020; Zhu and Woodcock, 2014b).
Leveraging the increasingly massive amount of openly available, analysis-ready data products
into the generation of operational land cover and land change information has been described as
the new paradigm for land cover science (Wulder et al., 2018). The approach, which intended to
use all available medium resolution remotely sensed data from the 1980s to the present, opened a
door for the scientific community to integrate time series information to improve change
detection and land cover characterization in a robust way. Furthermore, change events, when
combined with knowledge of ecology settings or anticipation of a given process post-change, can
accommodate consistent change observations and characterization of land cover. For example,
forest areas that are cleared by wildfire or harvest activities typically transfer to non-forest
herbaceous or shrub vegetation cover, followed by a succession of young tree stages, ultimately
returning to a forest class. Traditional change detection methods using limited observations may
not have identified these changes if data were collected with a starting date prior to the change
and an ending date that occurred after the transitional (non-tree) vegetation returned to tree
cover. Therefore, incorporating change information into the land cover characterization process
allows for insights regarding expected land cover class transitions related to successional
processes, and likewise provides a mechanism to identify illogical class transitions and cause or
agent of change  (Kennedy et al., 2015; Wulder et al., 2018). The choice of a time series
approach also allows missing data and phenological variations to be handled robustly (Friedl et
al., 2010; Wulder et al., 2018).
The Continuous Change Detection (CCD) and Classification (CCDC) algorithm (Zhu and
Woodcock, 2014b; Zhu et al., 2015b) was developed to advance time series change detection by
using all available Landsat data. The CCD algorithm uses robust methodology to identify when
and how the land surface changes through time. The algorithm first estimates a time series model
based on clear observations and then detects outliers by comparing model estimates and Landsat
observations. The algorithm fits harmonic regression models through a Least Absolute Shrinkage
and Selection Operator (LASSO) (Tibshirani, 1996) approach to every pixel over time to
estimate the time series model defined by sine and cosine functions. New Landsat records are
compared to predicted results, and if the observed data deviate beyond a set threshold for all
records within a moving window period, then a model break is produced. The parameters used to
fit the model are used as inputs for the cover classifier for land cover characterization.
The original implementation of CCDC was written in the MATLAB programming language and
had been implemented for a regional land cover change assessment in the eastern CONUS (Zhu
and Woodcock, 2014b). The algorithm includes the automation of change detection/classification
and can monitor changes for different land cover types. The implementation of CCDC into a
large geographic extent still encounters several challenges: the availability of Landsat records
and training datasets, the effectiveness of choosing good quality Landsat records, and the
robustness to characterize land cover and change across various land cover types and conditions.
In this paper, we outlined major efforts and challenges in the implementation of CCDC for the
U.S. Geological Survey (USGS) Land Change Monitoring, Assessment, and Projection
(LCMAP) initiative (Brown et al., 2020). LCMAP focuses on using CCD/CCDC with time series
Landsat records and other ancillary information to produce annual land cover and change
products from 1985 to the present for the United States. We focused on how LCMAP employed
every observation in a time series of U.S. Landsat Analysis Ready Data (ARD) (Dwyer et al.,
2018) over a long period starting with the 1980s to determine whether change occurred at any
given point in the observation record. The CCDC algorithm that was initially developed for
abrupt change detection on the land surface was modified through lessons learned from the
prototype test to include both gradual land cover transition and abrupt land change so that the
algorithm could be used in an operational setting with the goals of robust, repeatable, and
geographically consistent results (Brown et al., 2020). The algorithm was further used to classify
the pixel to indicate what land cover type(s) were observed before and after a detected change on
the land surface. Classification in LCMAP was modified to improve representativeness of
training data and reduce notable artifacts including misclassification of rare classes and dramatic
increase in the amount of training data. The CCDC algorithm has since been translated into an
open-source library as Python code. The full implementation joined the CCD Python library with
the classification methodology in combination with data delivery/processing services made
available through the LCMAP Information Warehouse and Data Store (IW+DS) and evolved as
a national operational monitoring system.

**2    Data Sources**
The CCDC algorithm utilizes all available Landsat observations including surface reflectance,
brightness temperature, and associated quality data to characterize the spectral responses of
every pixel through harmonic regression model fits. The model fits are then used to categorize
each pixel time series into temporal segments of stable periods and to estimate the dates at which
the spectral time-series data diverge from past responses or patterns. The outcomes of model fits
and other input data are then used for classification. The algorithm requires several input datasets
to perform both change detection and classification.
**2.1 Landsat observations**
U.S. Landsat ARD have been processed to a minimum set of requirements and organized into a
form that can be more directly and easily used for monitoring and assessing landscape change
with minimal additional user effort. Landsat ARD Collection 1 provides consistent radiometric
and geometric Landsat products across Landsat 4-5 Thematic Mapper (TM), Landsat 7 Enhanced
Thematic Mapper Plus (ETM+), and Landsat 8 Operational Land Imager (OLI) / Thermal
Infrared Sensor (TIRS) instruments for use in time series analysis (Dwyer et al., 2018). Landsat
ARD is organized in tiles, which are units of uniform dimension bounded by static corner points
in a defined grid system (Fig. 1). An ARD tile is currently defined as 5,000 x 5,000 30-meter (m)
pixels or 150 x 150-kilometer (km). To implement CCDC algorithms to produce LCMAP
Collection 1.0 land change products in CONUS, all available Landsat ARD records of surface
reflectance and brightness temperature from the 1980s to 2017 were required.
**2.2 Land cover and ancillary datasets**
The CCDC algorithm employs every observation in a time series of Landsat data to determine
whether change has occurred at any given time. The algorithm further classifies the time series to
indicate what land cover types were observed before and after a detected change and further to
generate LCMAP annual land cover products (Table 1). The land cover products are produced by
using training data from NLCD in 2001. NLCD provides Anderson Level II (Anderson, 1976)
land cover classification for CONUS and outlying areas (Homer et al., 2020). Spectral index and
change metrics between cloud-corrected Landsat mosaics are used, among other information, to
identify change pixels (Jin et al., 2013). These metrics allow NLCD to incorporate temporal and
spectral trajectory information into both training data selection and final land cover
classification. The NLCD land cover data is used in LCMAP as land cover training data.

Ancillary data comprises two main source datasets: the USGS National Elevation Dataset (NED)
(Gesch et al., 2002) 1 arc-second Digital Elevation Models (DEM), and a wetland potential index
(WPI) layer created for NLCD 2011 land cover production (Zhu et al., 2016). The WPI layer is a
ranking (0–8) of wetland likelihood from a comparison of the National Wetland Inventory
(NWI), the U.S. Department of Agriculture Soil Survey Geographic Database (SSURGO) for
hydric soils, and the NLCD 2006 wetlands land cover classes.

**3  Methodology**
As part of the operational LCMAP system, the original MATLAB version of the CCDC
algorithm is converted to a format that meets the needs of large-scale land change detection and
change characterization on an annual basis. Python is selected to replace MATLAB to implement
the CCDC algorithm for LCMAP.  The CCD component of the CCDC algorithm is converted to
create the Python-based CCD (PyCCD) library. The PyCCD library is a per-pixel algorithm, and
the fundamental outputs are the spectral characterizations (segments) of the input data. There are
several key components in PyCCD. The overall CCD procedures are summarized in Fig. 2.
**3.1 Data filtering and Harmonic modeling**
The removal of invalid and cloud-contaminated data points is important for deriving model
coefficients that accurately represent the phenology of the surface, and for the correct
identification of model break points. The CCD algorithm uses Landsat ARD PIXELQA values to
mask observations identified as cloud, cloud shadow, fill, or (in some cases) snow derived based
on the Fmask 3.3 algorithm (Zhu et al., 2015a; Zhu and Woodcock, 2012). Additional cirrus and
terrain occlusion bits are provided for Landsat 8 OLI-TIRS ARD that are not available in the
Landsat 4–7 TM/ETM+ quality assessment band. To maintain consistency across the historical
archive, the algorithm does not rely on these Landsat 8-only QA flags to filter out observations.
Landsat ARD containing invalid or physically unrealistic data values are removed. For the
surface reflectance bands, the valid data range is between 0 and 10000. Brightness temperature
values, which in the ARD are stored as $10 \times$ temperature (kelvin), are converted to $100 \times °C$ and
observations are filtered for values outside the range -9320 and 7070 (-93.2–70.7°C). This
procedure rescales the brightness temperature values into a roughly similar numerical range as
the surface reflectance bands. A multitemporal mask (Tmask) model (Zhu and Woodcock,
2014a) is implemented first to remove additional outliers by using the multitemporal observation
record to identify values that deviate from the overall phenology curve using a specific harmonic
model to perform an initial fit to the phenology. Additional details are provided in the
Supplementary materials S1.
The filtered Landsat ARD is further operated to generate the time series fit by harmonic models
whose sinusoidal components are frequency multiples of the base annual frequency. A constant
and linear term characterizes the surface reflectance or brightness temperature offset value and
overall slope, respectively. The full harmonic model is defined as follows:
$\hat{p}(i,t) = c_{0,i} + c_{1,i}t + \sum_{n=1}^{3}(a_{n,i}\cos\omega nt + b_{n,i}\sin\omega nt)$  (1)
where $\omega$ is the base annual frequency ($2\pi/T$), t is the ordinal of the date when January 1 of the
year zero has ordinal 1 (sometimes called Julian date), i is the ith Landsat band, $a_{n,i}$ and $b_{n,i}$ are
the estimated *nth* order harmonic coefficients for the ith Landsat band, $c_{0,i}$ and $c_{1,i}$ are the
estimated intercept and slope coefficients for the ith Landsat band, and $\hat{p}(i,t)$ is the predicted
value for the ith Landsat band at ordinal date t. Model initialization and certain special-case
regression fits such as at the beginning/end of the time series use the simple four-coefficient
model. Outside of these conditions, the selection of coefficient depends on the number of
observations used for the regression. For a full model (eight coefficients), there must be at least
24 observations covered by the regression. The fit parameters returned by PyCCD always
include eight coefficient values including an intercept, with unused coefficients reported as
zeroes.
**3.2 Regression models and change detection thresholds**
The best-fit coefficients for the time series model are calculated using a LASSO regression
model (Tibshirani, 1996). In contrast to Ordinary Least Squares (OLS) that was used in the
original CCDC development, LASSO penalizes the sum of the absolute values of coefficients, in
some cases forcing a subset of the coefficients to zero. Together with the explicit limits enforced
on the number of coefficients, this reduces instances of overfitting, including in cases when
observations are too sparse or unevenly distributed in time to constrain the model to real
phenological features. To detect change, the LASSO model checks CCD model breaks with
respect to its last determined best-fit harmonic model.
To correctly detect change, the algorithm distinguishes between a substantive deviation from
model prediction and deviations that result from variability inherent in the data (due to
incomplete atmospheric removal and/or other sources of natural variation) to detect change. The
algorithm calculates two parameters related to dispersion, or scatter, to estimate the variability of
data for each spectral band. The first one is a comparison root-mean-square-error (RMSE) that is
the RMSE of the 24 observations covered by the model which are closest in day of year to the
last observation in the "peek window," or over all observations covered by the model if there are
fewer than 24. This value is recalculated at each step of the time series. The second parameter
(*var*) is used to measure the overall variability of the data values and is defined as the median of
the absolute value of the differences between each observation and the ith successive
observation, where i is the smallest value such that the majority of these observation pairs are
separated by greater than 30 days, if possible (otherwise, i=1). The *var* is computed once at the
beginning of the standard procedure, using all non-masked observations in the time series.
Observations not yet incorporated into the model are evaluated as a group of no fewer than the
*PEEK_SIZE* parameter value; this is the "peek window," which "slides" along the time series
one observation at a time. Each iteration, a value is calculated for each individual observation
within the peek window, as follows:

$$mag_n = \sum_{i \in D} \left( \frac{resid_{n,i}}{max(var_i,\ RMSE_i)} \right)^2 \tag{2}$$

where, $resid_{n,i}$ is the residual relative to the LASSO models for each band $i$, for each
observation $n$ within the $PEEK\_SIZE$ window, $var_i$ and $RMSE_i$ are the parameters of dispersion
as described above, for each band $i$. This summation is carried out for all bands $i$ in the set of
$DETECTION\_BANDS$ ($D$). This produces a scalar magnitude, representing the deviation from
model prediction across these bands, for each observation. The detection of a model break
requires this value to be above the $CHANGE\_THRESHOLD$ value for all observations in the
window. This is separate from the value that is reported as a per-band magnitude when a change
is detected in the time series. Change detection sensitivity depends on the value of change
threshold. The $CHANGE\_THRESHOLD$ is determined in Eqs. S2 and S3 in the Supplementary.
If $mag_n < CHANGE\_THRESHOLD$ for any $n$ in the $Peek\_Size$ window, then add the most
recent observation to the segment by shifting the $Peek\_Size$ window one observation forward in
the time series. If $mag_n > CHANGE\_THRESHOLD$ for all $n$ in the $Peek\_Size$ window, this is
considered a spectral break.
**3.3 Permanent snow and insufficient clear observation procedures**
The permanent snow procedure indicates that too few clear (less than 25% of total observations)
or water observations, which are identified from the QA band, exist to robustly detect change,
and a large fraction of observations are snow. The algorithm will return at most one segment that
fits through the entire time series and provide the filtered observations number at least twelve.
The model will, under the default settings, fit only four coefficients (i.e., characterizing the
reflectance and brightness temperature bands using only a simple harmonic with no higher
frequency terms). Unlike other procedures, snow pixels are not filtered out and are fit as part of
the annual pattern. This avoids overfitting the model to a seasonally sparse observation record.
Similarly, for the insufficient clear observations determined by the QA band, the model will
perform a LASSO regression fit for the entire time series using four coefficients. The model
coefficients and RMSE from this regression are recorded. Additional parameters including the
start, end, and observation count are also saved. Further, the change Boolean value is set to 0,
and the break day is recorded as the last observation date. The magnitude of change as zero for
each band is also saved.

**3.4 Land cover classification**

The CCDC algorithm characterizes the land cover component of a pixel at any point using the
LCMAP time series model approach from the Landsat 4–8 records. The classification of CCDC
is accomplished for every pixel based on data from the time series models (e.g., model
coefficients). Land cover classifications are generated on an annual basis, using July 1st as a
representative date. A list of land cover classes and descriptions is provided in Table 1. Fig.3
illustrates an overall classification approach.

### 3.4.1    Classification algorithm

We chose eXtreme Gradient Boosting (XGBoost) (Chen and Guestrin, 2016) as the classification
method. XGBoost is a scalable implementation of gradient tree boosting, which is a supervised
learning method that can be used to develop a classification model when provided with an
appropriate training dataset. Generally, for a given dataset, a tree ensemble model uses additive
functions, which correspond to independent tree structures, to predict the land cover. The
predictions from all trees are also normalized to the final class probabilities using the softmax
function. The algorithm can handle sparse data and theoretically justify weighted quantile sketch
for approximate learning. The resultant trained model can be applied to a larger dataset to
generate predictions and probability scores which are the basis for LCMAP primary and
secondary land cover types. The primary and secondary land cover confidence values are
calculated from these scores.

### 3.4.2    Training dataset

The training data used in XGBoost for the LCMAP Collection 1.0 land cover products is from
the USGS NLCD 2001 land cover product (Homer et al., 2020). To meet the LCMAP land cover
legend, the NLCD data is first cross-walked to LCMAP classes, as shown in Fig.4 and Table 2.
The use of NLCD data that was cross-walked to the LCMAP land cover legend as the training
data will reduce uncertainties and improve the consistency of annual land cover change. For
example, grass and shrub have different ecological functions. Their spectral signatures are
distinct in some ecological regions but are very close in others, especially in the western
ecoregions of the conterminous United States (Underwood et al., 2007; Xian et al., 2013). Grass
and shrub usually grow close together, making it difficult to separate them in thematic land
cover. Combining these two cover classes can reduce uncertainties potentially caused by lack of
spectral distinction in Landsat observations. Furthermore, the extent of each land cover in the
cross-walked NLCD layer is eroded by one pixel. This step aims to reduce potential noise in the
classifier by removing pixels that may be heavily mixed with different cover types, or whose
land cover label may be less reliable. It also removes the narrow linear low-intensity developed
pixels corresponding to road networks, which were found to have registration issues with
Landsat ARD in some areas.

**3.4.3 Ancillary data**
Ancillary data used in the classification contains two main datasets: the DEM and the WPI layer.
Three DEM derivative datasets are implemented as geographic references for land cover
classification as ancillary data including topographic slope, aspect, and position index. The WPI
is highly related to wetland distribution and has a potential to improve wetland classification in
LCMAP.
**3.4.4 Classification procedures**
For each pixel, CCD segment data for the segment that includes the July 1st, 2001 date is used
with training data to create classification models (Zhou et al., 2020; Zhu et al., 2016). Data
generated from the CCD models are used to make the land cover classification because different
land cover classes can have different shapes for the estimated time series models. The
coefficients of the CCD models including the overall mean and model coefficients except
intercepts can be used to estimate the intra-annual changes caused by phenology and sun angle
differences for the ith Landsat band. The information obtained from the time series model is
useful for land cover classification. The CCD model data used with training data include the
model coefficients (except the intercepts) generated from surface reflectance and brightness
temperature bands, the model RMSE value for each band, and an average intercept value that is
calculated from average annual reflectance values for each band for the July 1, 2001 year. The
model training procedure is conducted at the tile level, using random samples drawn from the
targeted tile as well as the eight surrounding tiles to avoid not having enough training samples of
rare land cover types in the targeted tile. Cross-walked and eroded NLCD data are used for
classification labels, while the CCD model outputs and ancillary data are provided as
independent variables. Based on training data testing using different sample sizes, a target
sample size of 20 million pixels from the extent of 3x3 ARD tiles is chosen, requiring
approximately proportional representation of classes with the added constraint that no class be
represented by fewer than 600,000 or more than 8 million samples. If there are fewer than
600,000 samples available for a class, then all of the available samples are used without any
oversampling. The XGBoost hyperparameters are selected as maximum tree depth 8, fast
histogram optimized approximate greedy algorithm for tree method, multiclass logloss for
evaluation metric, and maximum number of rounds 500.
After the classification models in a given tile are trained, predictions are generated for each July
1st date that has an associated CCD segment (Fig. 5). The prediction information is supplied to
the production step for the creation of land cover. The process is repeated for each tile for the
entire CONUS ARD extent.
**3.5 Validation data**
The LCMAP land cover product is validated using an independent reference dataset. The
reference data, which consists of 24,971 30 m x 30 m pixels selected via a simple random
sampling method over CONUS, is collected from these sample plots between 1985 and 2017.
The TimeSync tool is used to efficiently display Landsat data for interpretation and to record
these interpretations into a database (Cohen et al., 2010; Pengra et al., 2020b; Stehman et al.,
2021). TimeSync displays the input Landsat images in two basic ways: by annual time-series
images and by pixel values plotted through time. For the image display, single 255 x 255-pixel
subsets of Landsat images in the growing season are displayed in sequence from 1984 to 2018.
Trained interpreters have access to all available images in each year to collect attributes in three
basic categories: 1) land use, 2) land cover, and 3) change processes. Additional attribute details
for the change processes, such as clear-cut and thinning associated with harvest events, are also
collected. The interpreters manually label these attributes using Landsat 5, 7, and 8 imagery,
high-resolution aerial photography, and other ancillary datasets (Cohen et al., 2010; Pengra et al.,
2020b). Interpreters also use ancillary data to support interpretation of Landsat and high-
resolution imagery, although Landsat data takes the highest weight of evidence. Recording the
full set of attributes in land use, land cover, and land change categories provides sufficient
information to meet the needs of LCMAP as well as other potential users. Quality assurance and
quality control (QA/QC) processes are also implemented to ensure the quality and consistency of
the reference data among interpreters and over the time span of data collection (Pengra et al.,
2020b). Each reference sample is interpreted by a trained interpreter and about 60% of these
pixels are interpreted independently by a second analyst.  Much of the QA/QC process relies on
comparing the interpretations at these duplicated sample pixels. Duplicated sample pixels that
have interpreter disagreement are evaluated in the QA/QC process, focusing on identifying
issues with specific classes or interpreters, flagging sample pixels for further review and possible
editing, and providing ongoing training and feedback to interpreters throughout the collection
process.  QA/QC related reviews are also completed on sample pixels that show interpretation
data such as uncommon and/or illogical land use and land cover combinations, multi-year
disturbance processes, rare classes, or other opportunistically identified situations.  Interpreted
attributes of sample pixels are edited, if necessary, to create the final attribute assignments for
the reference data.  These final attributes are then cross-walked to a single LCMAP land cover
class label, providing a single land cover reference label for each year of the time series for each
sample pixel.
The validation analysis protocols focus on estimating the confusion matrix and overall, user's,
and producer's accuracy by comparing the reference data and product data labels. Overall
accuracy and producer's accuracy as well as standard errors are produced using post stratified
estimators (Card, 1982; Stehman, 2013). For accuracy estimates that are produced by combining
multiple years of data, the sampling design is treated as a one-stage cluster sample where each
pixel represents a cluster and each year of observation is the secondary sampling unit using
cluster sampling standard error formulas (Pengra et al., 2020b; Stehman et al., 2021). The
validation is only performed for primary land cover and change products, not for other LCMAP
science products (Supplementary Section 4).
**3.6 Information warehouse and data store**
LCMAP adopts an information warehouse and data store (IW+DS) system that can expand
storage solutions along with data access and discovery services running on the EROS Shared
Mesos Cluster. The system provides different storage solutions to allow for flexibility in
choosing what best fits a dataset's characteristics and currently comprises Apache Cassandra
(https://cassandra.apache.org/ ) and Ceph ( https://ceph.io/ ) object storage. The services provide
data ingest, retrieval, discovery, metadata, processing, and other functionalities. LCMAP
maintains a copy of Landsat Collection 1 ARD and other similarly tiled ancillary datasets that
are spatially subset within the IW+DS to allow efficient retrieval and to enable large-scale
CCDC processing and other algorithmic work. The ingest process is designed to avoid bringing
in ARD tile observations that are already present within the IW+DS, to keep the input consistent
with any prior usage while allowing CCDC to bring in new observations as they are available.
Algorithmic results, products, and other intermediate data are kept on the Ceph object store
arranged using a prefix structure to label the identity of the data, with the actual object names
incorporating spatial concepts such as tile and chip that is a small subset of a tile and contains
100 by 100 30-m pixels.

**4    Results and Discussion**

The LCMAP primary land cover and change products were evaluated to outline annual land
cover change from 1985 to 2017 in the conterminous Unites States.
**4.1 Collection 1.0 primary land cover distribution and change**
The CONUS primary land cover mapping result and the primary confidence in 2010 are shown
in Fig. 6a and b, respectively. The land cover map illustrates distributions of different land cover
types across CONUS. The primary confidence is above 90% for most land cover classes,
suggesting that the classification models were created with high confidence for land cover
mapping for most classes in most regions. Some vegetation transition (green in Fig. 6b) occurs
mainly in the southeast region suggesting gradual tree recovery from disturbances associated
with tree harvesting. Fig. 6c and d display numbers of land cover changes and spectral changes
detected by the CCDC model between 1985 and 2017. The number of land cover changes
represents how many times land cover has changed from one type to another for a specific pixel.
However, the number of spectral changes denotes how many times the model has detected
spectral changes in a CCD time series model where spectral observations have diverged from the
model predictions. These changes could relate to a change in thematic land cover or might
represent more subtle conditional surface changes. The southeast region shows more frequent
land cover changes in the 33 years (Fig. 6c). The western part of CONUS, however, contains
more spectral changes than in the east (Fig.6d). The NLCD land change estimates also show
similar change patterns between 2001 and 2016 (Homer et al., 2020). The different spatial
patterns in the total number of land cover changes (Fig. 6c) and detected spectral changes (Fig.
6d) suggest that not all changes lead to land cover change (e.g., drought and precipitation-related
changes in vegetation or grassland fire). The large numbers of spectral change were mainly
detected in the southern grassland area.
Fig. 7 shows the temporal changes of areas for eight land cover classes from 1985 to 2017.
Among all classes, grass/shrub, tree cover, and cropland were dominant land cover types,
followed by wetland, water, developed, barren, and snow/ice. The land cover and change
datasets show that developed land has a consistent increasing trend with an 8.4% increase while
barren increased 9.1% between 1985 and 2017. Overall, the developed and barren areas
increased $2.58 \times 10^4$ km$^2$ and $8.56 \times 10^3$ km$^2$, respectively. Other land cover categories do not have
such increasing patterns. As for water, although fluctuating, it had a generally increasing trend.
The area of wetland had a rapid decrease before 2000, following a relatively steady though
fluctuating trend. Net wetland extent declined about 0.4% from 1985 to 2017. The grass/shrub
and tree cover classes both experienced consistent increasing trends before 2008 and 1995,
respectively, with areas reaching about $2.85 \times 10^6$ km$^2$ for grass/shrub and $2.14 \times 10^6$ km$^2$ for tree
in these two years. These two land covers gradually decreased since then. Tree cover declines
after 1996, showing a decreasing rate of 2.8% between 1985 and 2017. The cropland decreased
from 1985 to 2008 and quickly increased after that. By 2017, the area of cropland reached a
similar level of cropland area in 1988. Furthermore, most land cover changes are located in the
southeast region where many pixels change more than one time. The changes detected by the
CCD model suggest that landscape in the Midwest and west are more dynamic than in the east.
Many areas experience multiple disturbances although most of these changes do not result in
land cover transition.
The south ARD tile outlined in Fig. 6(a) covers the northern Dallas region, and the spatial
patterns of land cover and change are shown in more detail in Fig. 8. The land cover distributions
in the region show that urban land expands considerably from 1985 (Fig. 8a), to 1990 (Fig. 8b),
and to 2016 (Fig. 8c). The land conversion was primarily from cropland and grass/shrub to
developed land. Lake Ray Roberts was created in the late 1980s and captured in the land cover
map (Fig. 8b&c). The lake and urban conversion are also visible in the change count from 1985
to 2016 (Fig. 8g), which mainly show as blue, suggesting a one-time conversion. On the other
hand, there is almost no change in the urban center (Fig. 8g). Fig. 8 (d-f) shows high
classification confidence at the urban center, water, grass/shrub, and tree cover areas, whereas
cropland has relatively low confidence, indicating frequent management activities over croplands
in the regions. The total pixels of different change numbers suggest that one to two change times
are dominant, although some pixels change more than three times (Fig. 8h). The land cover
distributions in 1985, 1990, and 2017 show an increase in developed land and decreases in
cropland and grass/shrub (Fig. 8i).
The spatial patterns of land cover and change in the north ARD tile displayed in Fig. 6(a) in
northern Wyoming are shown in Fig. 9. The tile covers most of Yellowstone National Park, in
which tree, grass/shrub, and water are three dominant land cover types. Land cover in 1985,
1990, and 2016 (Fig. 9a-c) changed from tree to grass/shrub and back to tree cover. The primary
land cover confidence layers exhibit changes as decreasing vegetation from tree to grass/shrub
and increasing vegetation from grass/shrub to tree (Fig. 9d-f). For those trees and water bodies
that did not experience any disturbances, their magnitudes of confidence are relatively large. The
change map suggests that most forest lands experienced at least one change and some areas
changed multiple times (Fig. 9g). Most changes in forest lands were related to wildland fires that
occurred in the region. In 1988, 50 fires burned a mosaic covering nearly 3213 km$^2$ in
Yellowstone as a result of extremely warm, dry, and windy weather (NPS, 2021). Trees regrew
in some of the burn areas and these changes could occur more than once as shown in the change
map that indicates at least two changes in these areas. The total pixels of different change
frequencies suggest that one to two changes were dominant and very few pixels changed more
than three times (Fig. 9h). The land cover distributions in 1985, 1990, and 2017 had increases in
grass/shrub after 1985 and reductions in tree cover after that (Fig. 9i).
**4.2 Validation of land cover product**
The overall accuracy between the annual reference land cover label and the LCMAP annual land
cover products was calculated as 82.5% (±0.22%, standard error) when summarized for all years.
Overall accuracy across the time series (1985-2017) varied within about 1.5% annually, ranging
from a high of 83% in the late 1990s to about 82% in the late 2010s (Fig. 10). Per class
accuracies across CONUS ranged between 43% and 96% for user's accuracy (Table 3), with
water showing the highest accuracy (96% ±0.5% user's accuracy and 93% ±0.7% producer's
accuracy). Cropland has about 93% (±0.3%) producer's accuracy and 70% (±0.6%) user's
accuracy. The lowest accuracies are observed for barren and wetland. The per class per year
agreements show the accuracies vary slightly for each class in each year (Table 4). The
variations of annual overall accuracy are within a range of about 1.5% across the time series. The
slight decline in annual overall accuracy suggests that year-to-year trends may be a result of a
complex interplay of temporal biases in the LCMAP algorithm, Landsat data quality and
quantity, the model break detection accuracy of the LCMAP CCD, and errors in the training data
used for the classification. For example, the change detection portion of the algorithm is known
to be conservative in identifying land cover change. The CCD model assumes that the spectral
variations of the land surface through time can be characterized with annual harmonic models
and can be separated into discrete periods of time. Therefore, the model performs better when the
short-term spectral variability of the land surface is low, the changes have a large spectral
response, and the observational data density is high. Over time, the actual land cover may evolve
away from the phenology represented by spectral models that may have missed one or more
spectral breaks, which will impact accuracy especially when the land cover changes are
persistent rather than cyclic, such as with an expanding urban footprint. Annual accuracy of
Developed showed an upward trend in user accuracy (UA) and a downward trend in producer
accuracy (PA) over time (Stehman et al., 2021). The increasing availability of high-resolution
data used by the interpreters may have increased the likelihood of identifying features
characteristic of Developed land that could not be identified earlier in the time series, leading to
an increase in the proportion of Developed area estimated from the sample. Consequently, the
increasing sensitivity of the reference interpretation to landscape features may account for the
difference between the mapping and the reference data over time. Lower data density toward the
beginning and end of the time series may decrease accuracy, which when combined with other
factors, can contribute to the annual land cover overall accuracy across all years.
**4.3 Significance of the product**
One of the biggest advances of LCMAP relative to conventional methods available to date is its
approach of generating annual land change products by using the entire Landsat archive at a
large geographic scale. Landsat ARD, which is the foundation for LCMAP, is effective and
straightforward for tracking and characterizing the historical land changes at a pixel level over
decades. Compared to conventional methods, detecting changes using all available observations
enables us to date these changes as they occur. After change is detected, temporally consistent
land cover products rather than stochastic changes in labels can be produced at annual intervals
by conducting classification from CCD model segmented contributions
The LCMAP product suite includes five land cover change and five land surface change science
products. It represents a new paradigm that consistently and continuously provides a large
volume of land change information for land change monitoring, land resource management, and
scientific research. In addition to primary and secondary land cover before and after changes,
change segments containing spectral change time and magnitude are provided to explore the
changes in land condition and could meet various user communities' needs. The LCMAP
products can improve our understanding of causes, rates, and consequences of the land surface
changes (Rover et al. 2020) such as forest changes caused by wildfire and insect outbreaks.
By implementing the CCDC algorithm through a system engineering approach, LCMAP
provides a fully automated framework for land change monitoring. The framework can also be
updated to include the latest Landsat records so that it can be used for operational continuous
monitoring in a large geographic extent (Brown et al. 2020). Therefore, when new observations
become available, the framework can provide timely and consistent land cover characteristics to
the public.
**4.4 Limitations and challenges**
Although LCMAP Collection 1.0 products have been proven to be successful in detecting
various land surface changes to support research applications related to environment and ecology
conditions, limitations and challenges exist. Utilizing Landsat ARD data as input provided
consistent time series Landsat imagery with high level geometric and radiometric quality for
implementing the CCDC method. Nevertheless, the densities of Landsat observation records
varied greatly across space and time due to spatial differences in Landsat scene overlap and
temporal coverage, as well as regional differences in contamination by clouds, cloud shadows,
and snow. The change detection accuracies of CCD models were highly influenced by the
temporal frequency of available observations. Zhou et al. (2019) found that using harmonized
Landsat-8 and Sentinel-2 (HLS) data increased the temporal frequency of the data and thus
enhanced the ability to model seasonal variation and derived better change detection results than
using Landsat data alone. Integrating multi-mission data could provide the opportunity to
enhance change detection, especially for the land cover types that are highly dynamic or in
frequently cloudy/snowy areas.
Providing only eight general land cover classes and their changes in LCMAP Collection 1.0
products limits the usage of the product in some applications that need a higher level of thematic
land cover detail. For example, shrub and grass are two major vegetation types and have
different ecological functions, but they are not delineated separately in LCMAP Collection 1.0
products. Lack of measurement of grassland-shrub transition constrains the study of shrub
encroachment, which is a symptom of land degradation. However, NLCD 2001 level I land
cover product had different mapping accuracies for different land cover types in different
ecological regions (Wickham et al., 2010). For example, the grass mapping accuracies were
higher in the eastern regions than they were in most western mapping regions. The accuracies of
shrub cover had similar variation patterns across CONUS. These accuracy variations suggest
uncertainties of the products, especially in most western regions where grass and shrub are more
difficult to be separated. Combining grass and shrub from the NLCD 2001 product reduced
uncertainties introduced by the two individual components and made the accuracy of the
grass/shrub product in LCMAP relatively high and consistent across CONUS (Stehman et al.,
2021). NLCD has established new efforts to improve mapping accuracies by adding innovative
approaches for land cover classification and introducing continuous rangeland products in
western CONUS for NLCD thematic land cover products since 2001 (Homer et al., 2020). The
use of new NLCD products as the training data will support LCMAP to produce more land cover
types including separating grass and shrub in the future.
Adopting NLCD land cover product as the training data source efficiently provided abundant
training samples to deliver land cover product with high classification accuracy. Selecting a
sufficient size of training samples is important for CCDC models to obtain accurate
classification. Previous land cover post-classification analysis suggested that the overall
classification accuracy increased when the training samples increased (Gong et al., 2020). The
recent global land cover classification also suggested that the appropriate training sample size for
a mapping extent of three 158 km x 158 km tiles should be larger than 60,000 (Zhang et al.,
2021). For the LCMAP land cover classification, a much larger training size was utilized to
ensure that these training samples could represent landscape features in the classification tiles.
However, these training data were randomly selected from the NLCD land cover product,
suggesting errors could potentially be carried over to the training samples due to potential errors
in the training source. Besides uncertainties in training data, some obvious challenges such as
class definitional differences between pasture/hay and grassland between NLCD and LCMAP
could potentially be carried over to the LCMAP land cover product. Improving training data by
reducing uncertainties and potential errors in a more consistent and accurate way is critical to
strengthen land cover classification and to improve the scientific quality of LCMAP products in
the future.
There are apparent shifts in some land cover types, especially in snow/ice and barren (Fig.7), and
a decline in overall agreement (Fig.10) in 2017, the last year of the Collection 1.0 product. The
last year's product usually is provisional because limited Landsat observations are available at
the end of a time series. The CCDC requires at least 24 clear observations to create full models
for change detection and classification. Without sufficient clear observations, the algorithm
could not produce model break accurately. Therefore, in the last year of a time series, the rule-
based assignment is implemented to label land cover for these pixels that do not have enough
observations to build a time series model. Both primary and secondary land cover classes are
assigned from the last identified primary and secondary classes.

**5   Data Availability**
The LCMAP products generated in this paper are available at https://earthexplorer.usgs.gov/
(LCMAP, 2021). All LCMAP land change products are mosaiced for the conterminous United
States in the GeoTIFF format. Find exact data as described here at
https://doi.org/10.5066/P9W1TO6E. The reference dataset used for the product validation is also
available at https://www.sciencebase.gov/catalog/item/5e57e965e4b01d50924a93f6
or  https://doi.org/10.5066/P98EC5XR (Pengra et al., 2020a).

## 6 Conclusions

The continuous Landsat observations spanning from the 1980s to the present, new generations of
change detection and classification models, and systems capable of processing large volume data
are offering unprecedented opportunities to characterize land cover and detect land surface
change consistently and accurately. Additionally, the collection of reference data used to validate
land cover products provides validation result for each land cover category annually. To capture
the variability of landscape condition and its responses to different disturbances, land cover and
land surface change datasets need to be produced over a large geographic scale. LCMAP has
produced a suite of land change product at 30-m resolution including the reference dataset in the
United States. In that context, LCMAP was developed to generate an essential dataset to meet
broad scientific research and resource management needs. Using the CCDC algorithm and
Landsat ARD to determine whether change has occurred at any given point in the observation
record, LCMAP produced annual land cover and change datasets for the conterminous United
States in a robust manner. These new datasets and the novel production systems will allow for
new generation of research and applications in connecting time series remote sensing
observations with land surface change at a much finer scale than previously possible.

**Supplement.** The supplement related to this article is attached.

**Author contributions.**

KS conducted PyCCD programming for CCD/CCDC models. ZZ developed the original
MATLAB version of CCD/CCDC programs. JH participated in reference data collection. DW
and QZ assisted in data integration tasks. GX analysed the data and wrote the manuscript with
contributions from all co-authors.

**Completing interests**. The authors declare that they have no conflict of interest.

**Acknowledgements.**

Any use of trade, firm, or product names is for descriptive purposes only and does not imply
endorsement by the U.S. Government. Qiang Zhou and Congcong Li's work were performed
under Work performed under USGS contract 140G0119C0001.

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

**Caption of Table**

Table 1 LCMAP land cover product specifications

Table 2 NLCD land cover cross-walked to LCMAP land cover

Table 3. Confusion matrix for CONUS (all years combined) where cell entries represent percent of CONUS area. Overall accuracy is 82.5% (±0.22%). Standard errors for user's and producer's accuracies are shown in parentheses and $n$ is the number of sample pixels for each row and column.

Table 4 Overall per class agreement in percentage between 1985 and 2017

**Caption of Figure**

Figure 1 Landsat ARD tile grids for the conterminous U.S.

Figure 2 Overall procedures of the CCD algorithm.

Figure 3 The overall approach of land cover classification in CCDC.

Figure 4. NLCD 2001 land cover (a), cross-walked LCMAP land cover classes (b), LCMAP land cover eroded by one pixel (c), zoomed in cross-walked land cover from NLCD 2001 (d), and zoomed in LCMAP land cover classes eroded by one pixel (e). The color legends represent NLCD land cover class and LCMAP primary land cover (LCPRI).

Figure 5 CCD change detection and segmentation using Landsat blue, green, red, near-infrared, short-wave infrared (SWIR) 1, short-wave infrared (SWIR) 2, and thermal bands. Blue dots are all available clear Landsat records in each year. The horizontal lines in different colors represent land cover classes labeled by the algorithm. The vertical lines show model break dates. The back line is the model fits. The high-resolution images show landscape conditions in 2007 and 2013.

Figure 6 Illustration of the LCMAP product: (a) Primary land cover in 2010, (b) Primary land cover confidence in 2010, (c) the frequency of land cover changes from 1985 to 2017, and (d) total number of spectral changes detected from 1985 to 2017.

Figure 7 Areal variations of eight primary land cover types from 1985 to 2017 in CONUS.

Figure 8 Primary land cover and confidences in 1985 (a) and (d), 1990 (b) and (e), 2016(c) and (f), change in 1985-2017 (g), the frequency of land cover change (x-axis) from 1985 to 2017 and numbers of total pixels (y-axis) of these changes of different change (h), and areas (y-axis) of different land cover (x-axis) in the three times for the ARD tile 16_14 (i).

Figure 9 Primary land cover and confidences in 1985 (a) and (d), 1990 (b) and (e), 2016 (c) and (f), and change in 1985-2017 (g), the frequency of land cover change (x-axis) from 1985 to 2017 and numbers of pixels (y-axis) of these changes (h), and areas (y-axis) of different land cover (x-axis) in the three times for the ARD tile 9_6 (i).

Figure 10 Overall agreement between LCMAP primary land cover and reference data across CONUS. The cross lines represent +/- one standard errors.

Table 1 LCMAP land cover product specifications

| Code | Land Cover Class | Description |
|---|---|---|
| 1 | **Developed** | Areas of intensive use with much of the land covered with structures (e.g., high-density residential, commercial, industrial, mining, or transportation), or less intensive uses where the land cover matrix includes vegetation, bare ground, and structures (e.g., low-density residential, recreational facilities, cemeteries, transportation/utility corridors, etc.), including any land functionality related to the developed or built-up activity. |
| 2 | **Cropland** | Land in either a vegetated or unvegetated state used in production of food, fiber, and fuels. This includes cultivated and uncultivated croplands, hay lands, orchards, vineyards, and confined livestock operations. Forest plantations are considered as forests or woodlands (Tree Cover class) regardless of the use of the wood products. |
| 3 | **Grass/Shrub** | Land predominantly covered with shrubs and perennial or annual natural and domesticated grasses (e.g. pasture), forbs, or other forms of herbaceous vegetation. The grass and shrub cover must comprise at least 10% of the area and tree cover is less than 10% of the area. |
| 4 | **Tree Cover** | Tree-covered land where the tree cover density is greater than 10%. Cleared or harvested trees (i.e. clearcuts) will be mapped according to current cover (e.g. Barren, Grass/Shrub). |
| 5 | **Water Bodies** | Areas covered with water, such as streams, canals, lakes, reservoirs, bays, or oceans. |
| 6 | **Wetland** | Lands where water saturation is the determining factor in soil characteristics, vegetation types, and animal communities. Wetlands are composed of mosaics of water, bare soil, and herbaceous or wooded vegetated cover. |
| 7 | **Ice and Snow** | Land where accumulated snow and ice does not completely melt during the summer period (i.e. perennial ice/snow). |
| 8 | **Barren** | Land comprised of natural occurrences of soils, sand, or rocks where less than 10% of the area is vegetated. |

Table 2 NLCD land cover cross-walked to LCMAP land cover

| NLCD Value | LCMAP Value |
|---|---|
| Water | Water |
| Ice/Snow | Ice and Snow |
| Developed, open space; Developed, low intensity; Developed medium intensity; Developed, high intensity | Developed |
| Barren | Barren |
| Deciduous forest, Evergreen forest, Mixed forest | Tree Cover |
| Shrub/Scrub, Grassland/Herbaceous | Grass/Shrub |
| Hay/Pasture, Cultivated crops | Cropland |
| Woody wetland, Emergent herbaceous wetland | Wetland |

Table 3. Confusion matrix for CONUS (all years combined) where cell entries represent percent of CONUS area. Overall accuracy is 82.5% (±0.22%). Standard errors for user's and producer's accuracies are shown in parentheses and *n* is the number of sample pixels for each row and column.

| Map | Devel. | Crop. | Grass /Shrub | Tree | Water | Wetland | Ice/ Snow | Barren | Total | User (SE) | *n* |
|---|---|---|---|---|---|---|---|---|---|---|---|
| Devel. | **3.000** | 0.139 | 0.321 | 0.377 | 0.024 | 0.035 | | 0.001 | 3.896 | 77 (1.2) | 32102 |
| Crop. | 0.918 | **16.527** | 5.061 | 0.799 | 0.027 | 0.368 | | 0.003 | 23.702 | 70 (0.6) | 195283 |
| Grass /Shrub | 0.368 | 0.757 | **30.649** | 2.599 | 0.045 | 0.229 | | 0.332 | 34.980 | 88 (0.3) | 288197 |
| Tree | 0.340 | 0.143 | 1.414 | **23.387** | 0.049 | 0.579 | | 0.006 | 25.917 | 90 (0.3) | 213531 |
| Water | 0.013 | 0.008 | 0.048 | 0.024 | **4.788** | 0.067 | | 0.020 | 4.968 | 96 (0.5) | 40932 |
| Wetland | 0.062 | 0.129 | 0.361 | 0.944 | 0.172 | **3.688** | | 0.001 | 5.357 | 69 (1.3) | 44136 |
| Ice/Snow | | | 0.004 | 0.004 | | 0.004 | **0.012** | 0.004 | 0.028 | 43 (18.7) | 231 |
| Barren | 0.072 | 0.005 | 0.501 | 0.013 | 0.056 | 0.012 | | **0.492** | 1.151 | 43 (2.8) | 9485 |
| Total | 4.772 | 17.707 | 38.358 | 28.149 | 5.162 | 4.981 | 0.012 | 0.859 | 100.00 | | |
| Prod (SE) | 63 (1.3) | 93 (0.3) | 80 (0.4) | 83 (0.4) | 93 (0.7) | 74 (1.2) | 100 (0) | 57 (3.2) | | | |
| *n* | 39319 | 145886 | 316027 | 231916 | 42530 | 41042 | 99 | 7078 | | | |

Table 4 Overall per class agreement in percentage between 1985 and 2017

| Overall Per Class Agreement | Developed | Cropland | Grass/Shrub | Tree | Water | Wetland | Snow/Ice | Barren |
|---|---|---|---|---|---|---|---|---|
| 1985 | 66 | 80 | 83 | 87 | 95 | 72 | 60 | 49 |
| 1986 | 67 | 80 | 83 | 87 | 95 | 72 | 60 | 49 |
| 1987 | 68 | 80 | 83 | 86 | 95 | 72 | 60 | 49 |
| 1988 | 68 | 80 | 83 | 87 | 95 | 72 | 60 | 49 |
| 1989 | 68 | 80 | 84 | 87 | 95 | 72 | 60 | 48 |
| 1990 | 68 | 80 | 84 | 87 | 95 | 72 | 60 | 48 |
| 1991 | 68 | 80 | 84 | 87 | 95 | 72 | 60 | 49 |
| 1992 | 69 | 80 | 84 | 87 | 95 | 71 | 60 | 50 |
| 1993 | 69 | 80 | 84 | 87 | 95 | 71 | 60 | 49 |
| 1994 | 69 | 80 | 84 | 87 | 95 | 71 | 60 | 49 |
| 1995 | 70 | 80 | 84 | 87 | 95 | 72 | 60 | 49 |
| 1996 | 69 | 80 | 84 | 87 | 95 | 72 | 60 | 48 |
| 1997 | 70 | 80 | 84 | 87 | 95 | 72 | 60 | 49 |
| 1998 | 70 | 80 | 84 | 87 | 94 | 72 | 60 | 48 |
| 1999 | 70 | 80 | 84 | 87 | 95 | 72 | 60 | 48 |
| 2000 | 70 | 80 | 84 | 87 | 95 | 72 | 60 | 48 |
| 2001 | 70 | 80 | 84 | 87 | 95 | 72 | 60 | 49 |
| 2002 | 70 | 80 | 84 | 86 | 95 | 72 | 60 | 49 |
| 2003 | 70 | 80 | 84 | 87 | 94 | 71 | 60 | 48 |
| 2004 | 69 | 80 | 84 | 86 | 94 | 71 | 60 | 48 |
| 2005 | 70 | 80 | 84 | 86 | 94 | 71 | 60 | 49 |
| 2006 | 70 | 79 | 84 | 86 | 94 | 71 | 60 | 49 |
| 2007 | 70 | 79 | 84 | 86 | 94 | 71 | 60 | 50 |
| 2008 | 70 | 79 | 84 | 86 | 94 | 71 | 60 | 49 |
| 2009 | 70 | 79 | 84 | 86 | 94 | 71 | 60 | 49 |
| 2010 | 70 | 79 | 84 | 86 | 94 | 71 | 60 | 50 |
| 2011 | 70 | 79 | 84 | 86 | 94 | 71 | 60 | 51 |
| 2012 | 70 | 79 | 83 | 86 | 94 | 71 | 60 | 50 |
| 2013 | 69 | 79 | 83 | 86 | 94 | 71 | 60 | 50 |
| 2014 | 69 | 79 | 83 | 86 | 94 | 71 | 60 | 50 |
| 2015 | 69 | 79 | 83 | 86 | 94 | 71 | 60 | 50 |
| 2016 | 69 | 79 | 83 | 86 | 94 | 71 | 60 | 50 |
| 2017 | 69 | 78 | 83 | 85 | 94 | 70 | 60 | 49 |

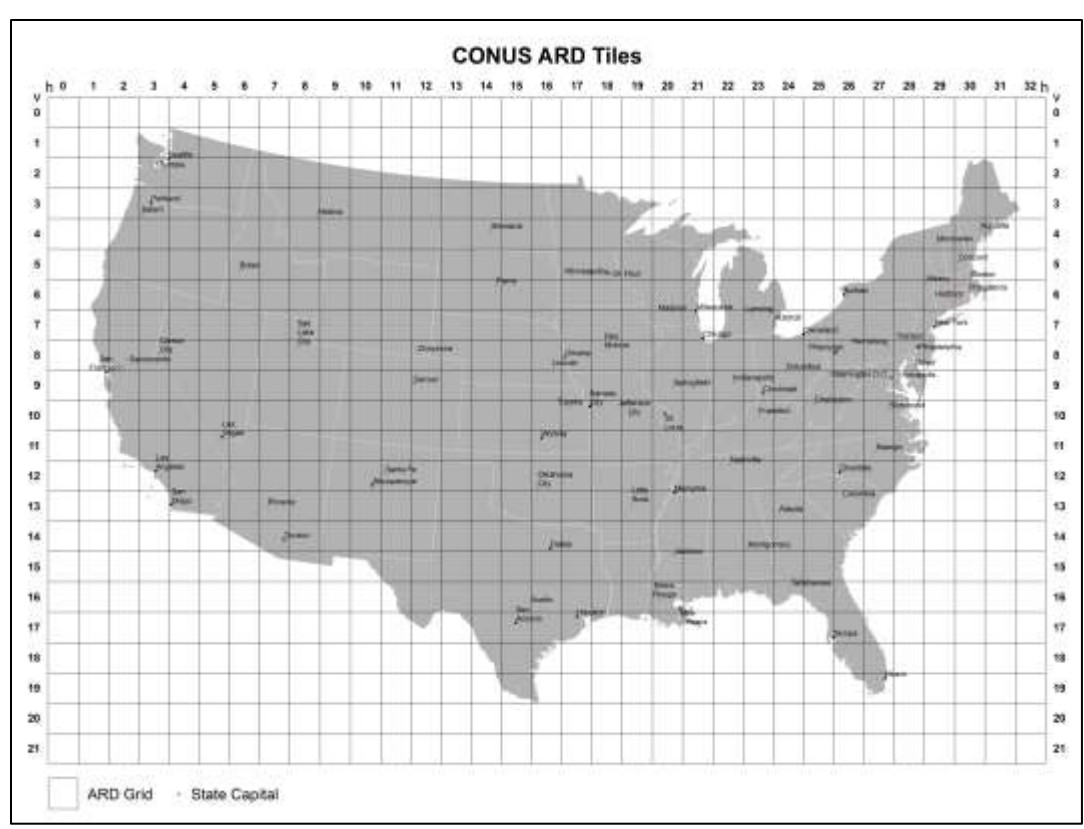

Figure 1 Landsat ARD tile grids for the conterminous U.S.

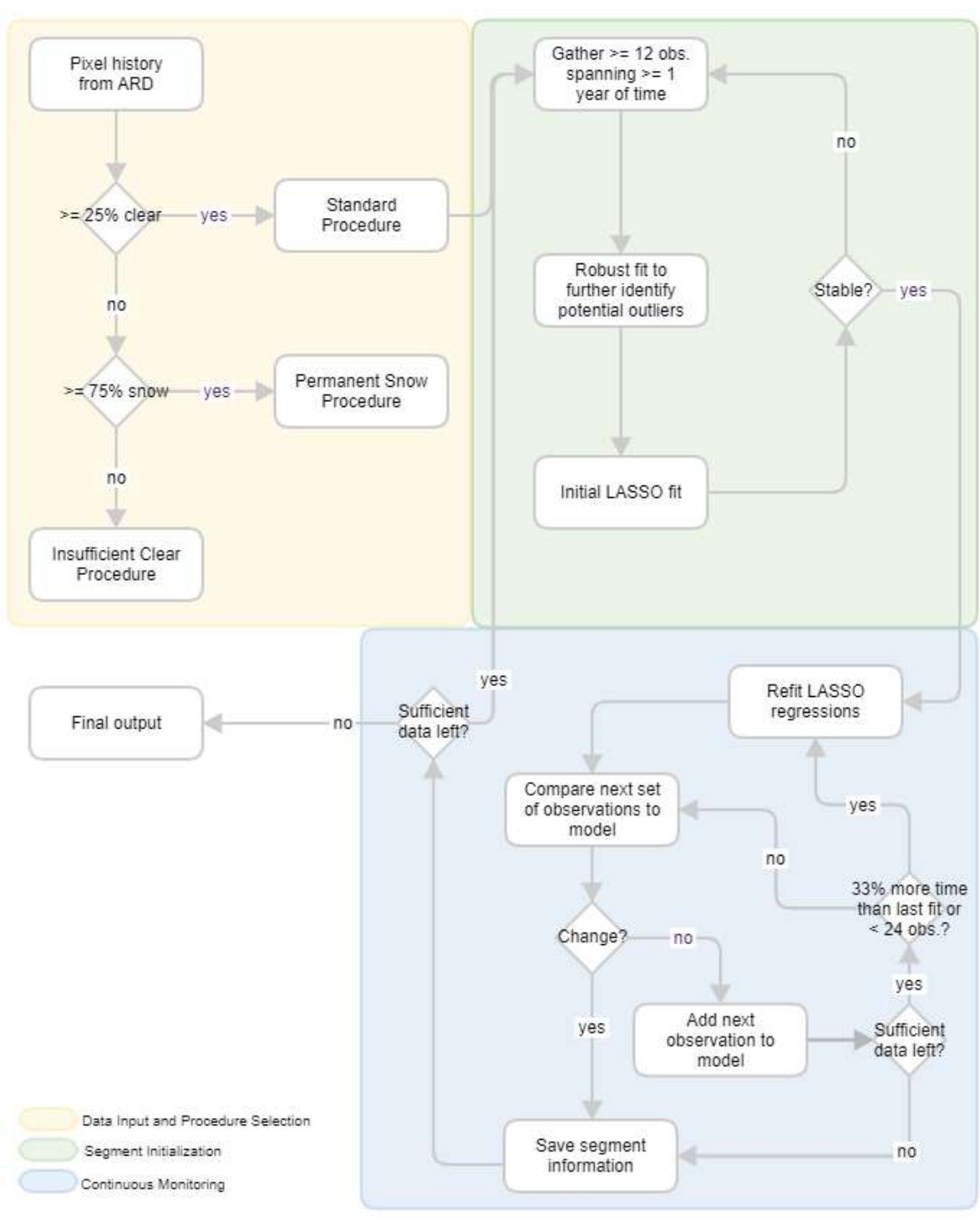

Figure 2 Overall procedures of the CCD algorithm.

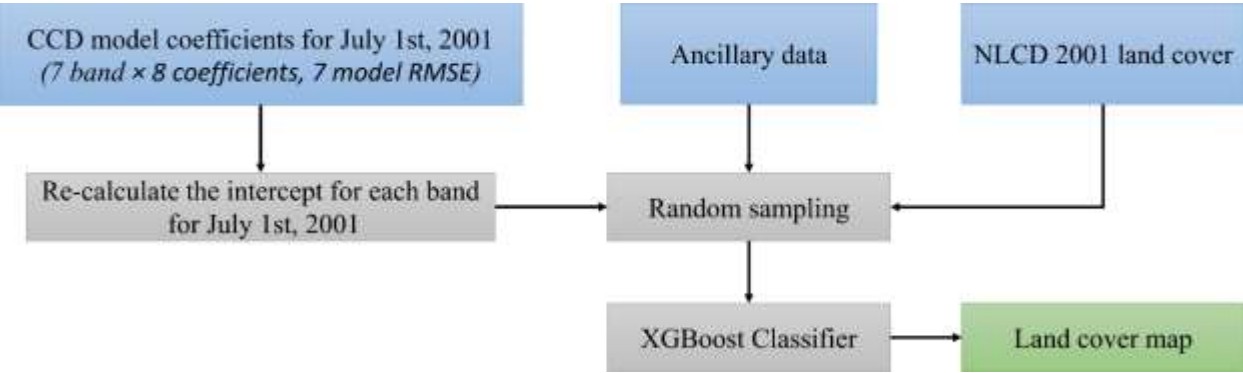

Figure 3 The overall approach of land cover classification in CCDC.

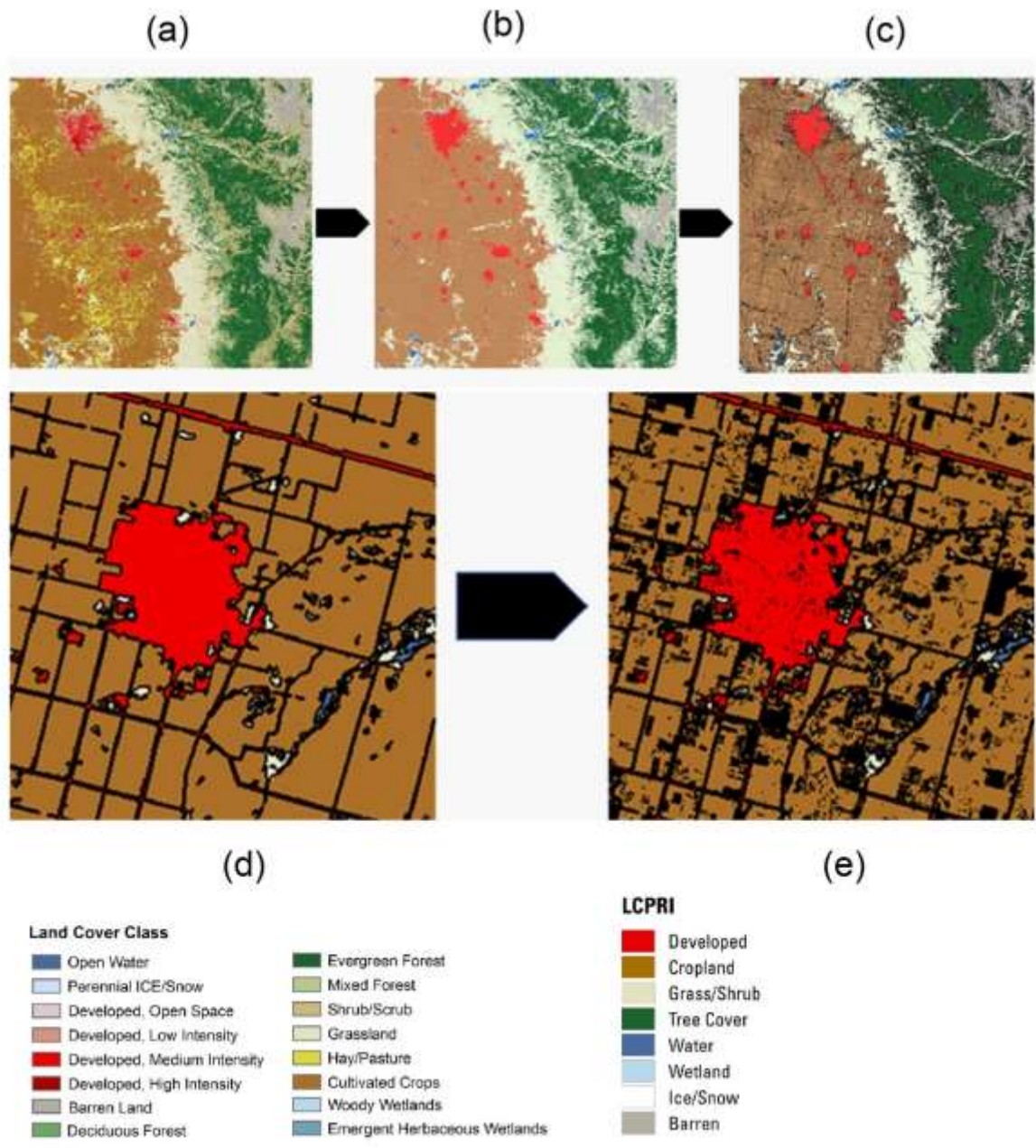

Figure 4. NLCD 2001 land cover (a), cross-walked LCMAP land cover classes (b), LCMAP land cover eroded by one pixel (c), zoomed in cross-walked land cover from NLCD 2001 (d), and zoomed in LCMAP land cover classes eroded by one pixel (e). The color legends represent NLCD land cover class and LCMAP primary land cover (LCPRI).

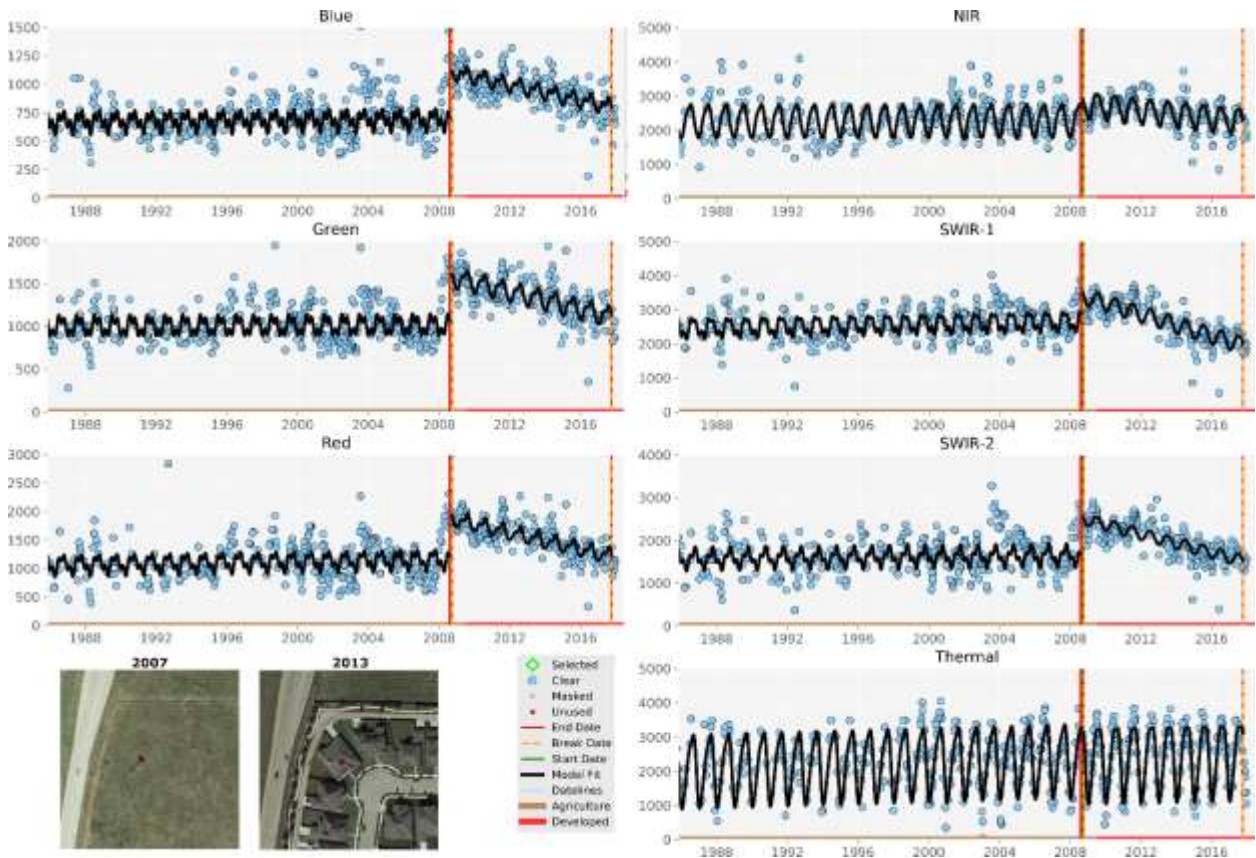

Figure 5 CCD change detection and segmentation using Landsat blue, green, red, near-infrared, short-wave infrared (SWIR) 1, short-wave infrared (SWIR) 2, and thermal bands. Blue dots are all available clear Landsat records in each year. The horizontal lines in different colors represent land cover classes labeled by the algorithm. The vertical lines show model break dates. The back line is the model fits. The high-resolution images show landscape conditions in 2007 and 2013.

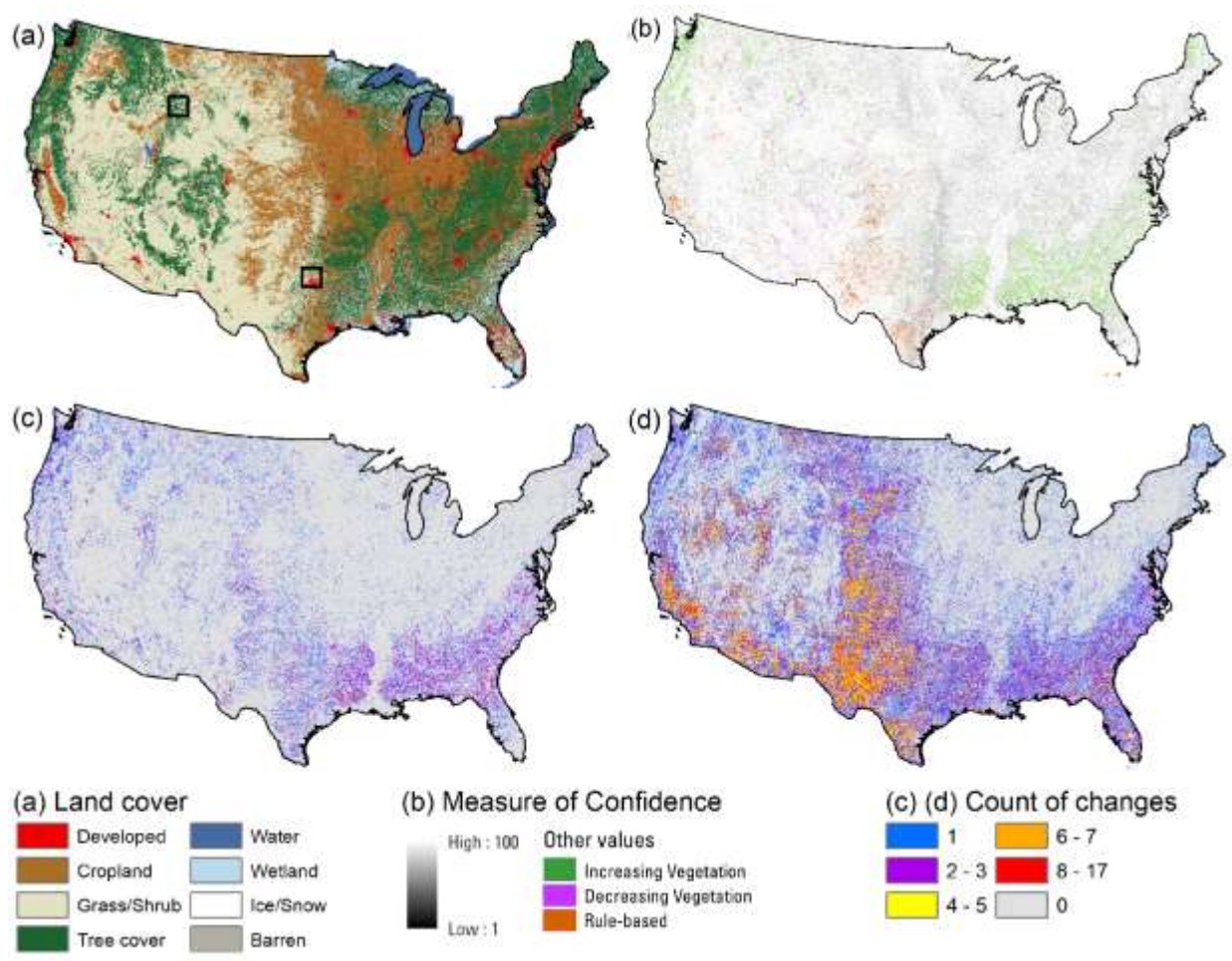

Figure 6 Illustration of the LCMAP product: (a) Primary land cover in 2010, (b) Primary land cover confidence in 2010, (c) the frequency of land cover changes from 1985 to 2017, and (d) total number of spectral changes detected from 1985 to 2017.

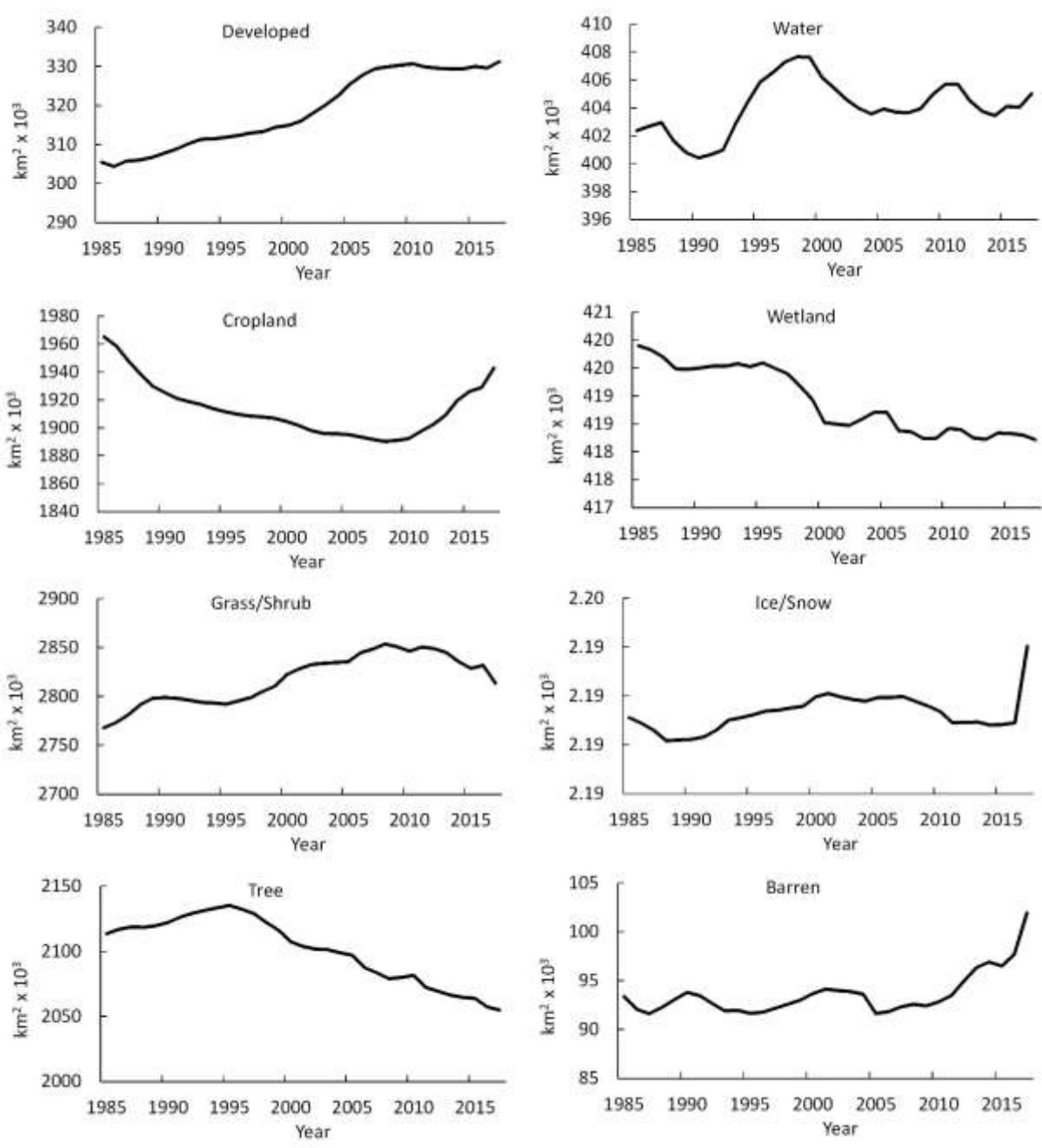

Figure 7 Areal variations of eight primary land cover types from 1985 to 2017 in CONUS.

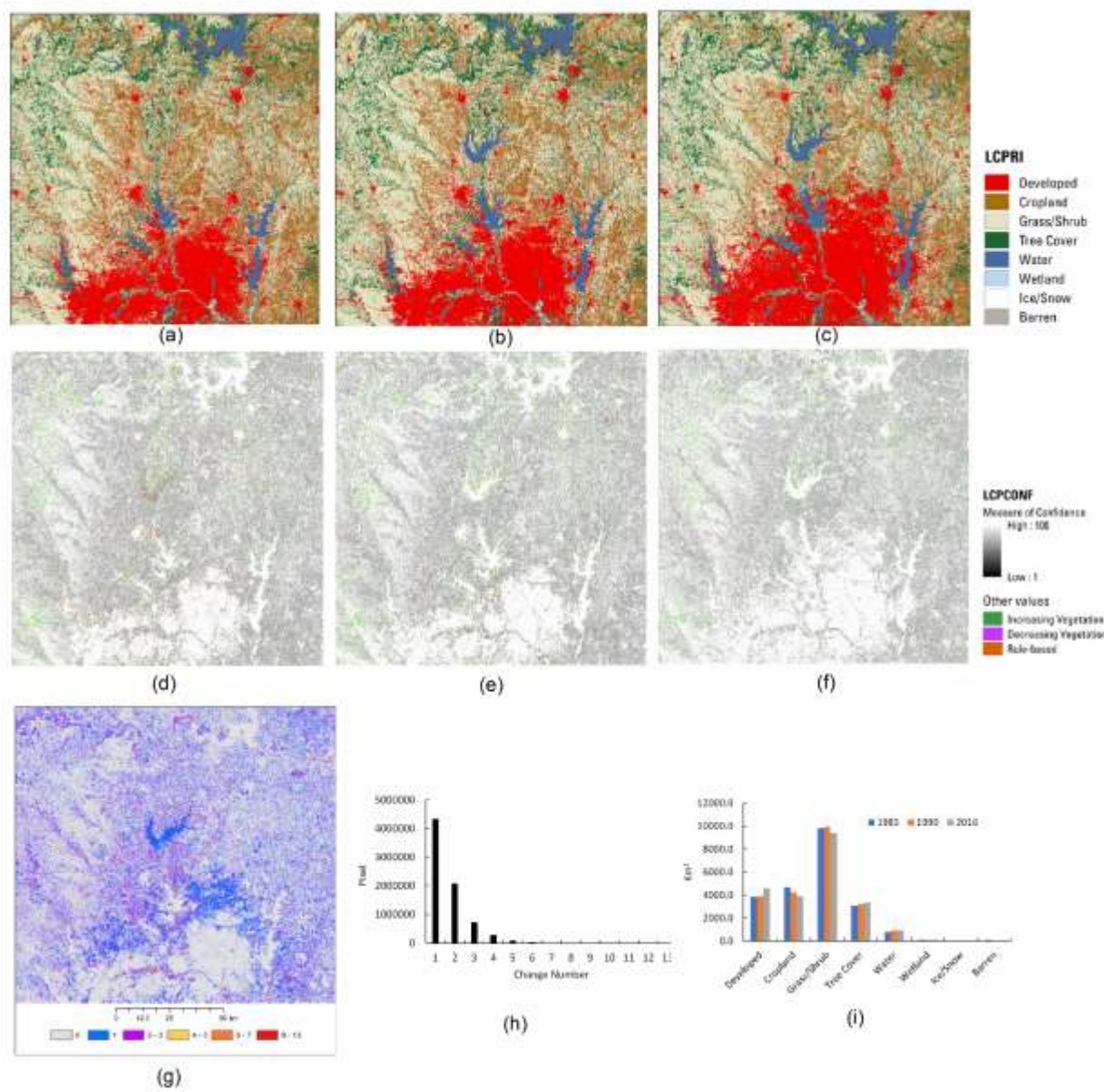

Figure 8 Primary land cover and confidences in 1985 (a) and (d), 1990 (b) and (e), 2016(c) and (f), change in 1985-2017 (g), the frequency of land cover change (x-axis) from 1985 to 2017 and numbers of pixels (y-axis) of these changes (h), and areas (y-axis) of different land cover (x-axis) in the three times for the ARD tile 16_14 (i).

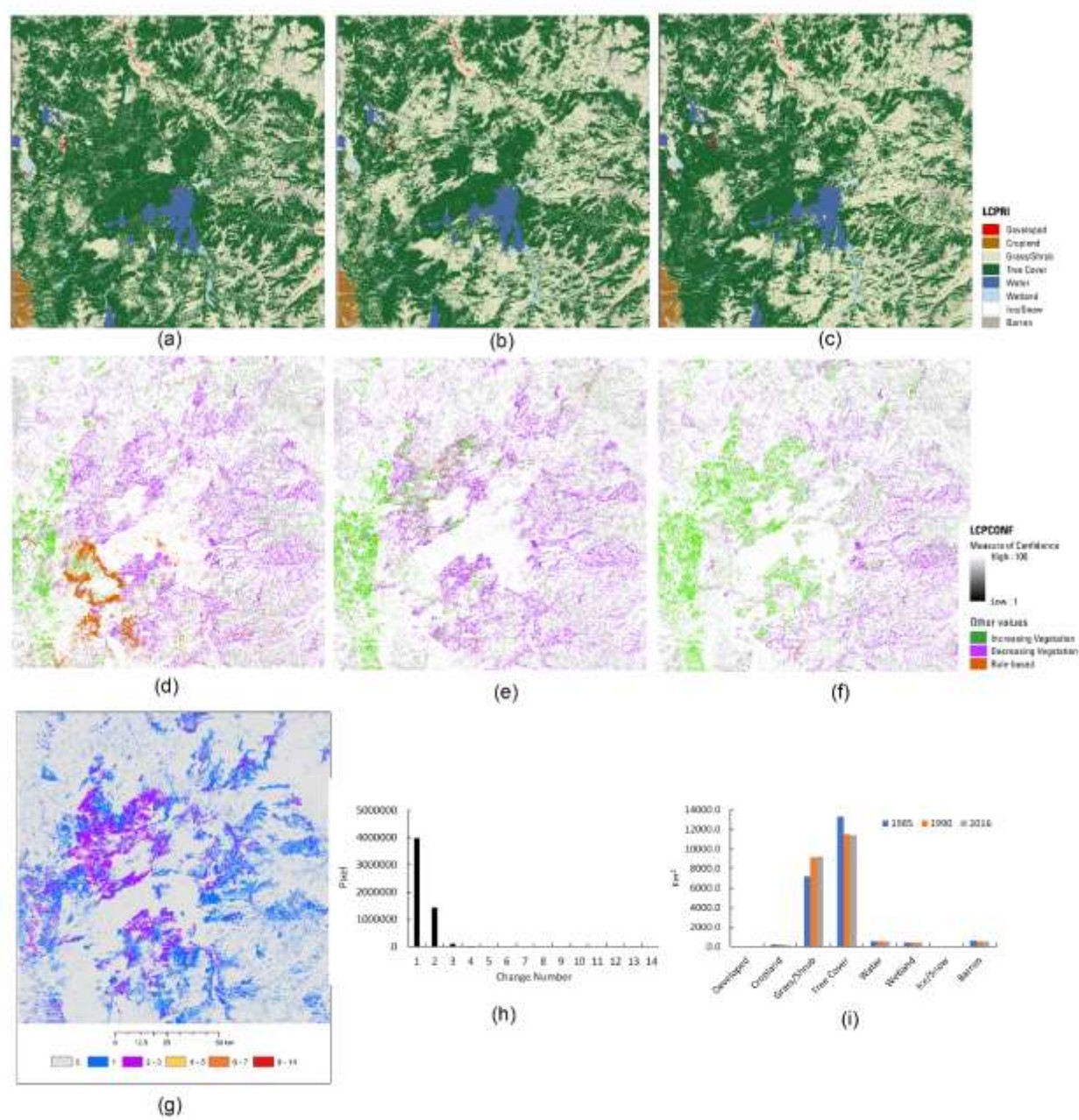

Figure 9 Primary land cover and confidences in 1985 (a) and (d), 1990 (b) and (e), 2016 (c) and (f), and change in 1985-2017 (g), the frequency of land cover change (x-axis) from 1985 to 2017 and numbers of pixels (y-axis) of these changes (h), and areas (y-axis) of different land cover (x-axis) in the three times for the ARD tile 9_6 (i).

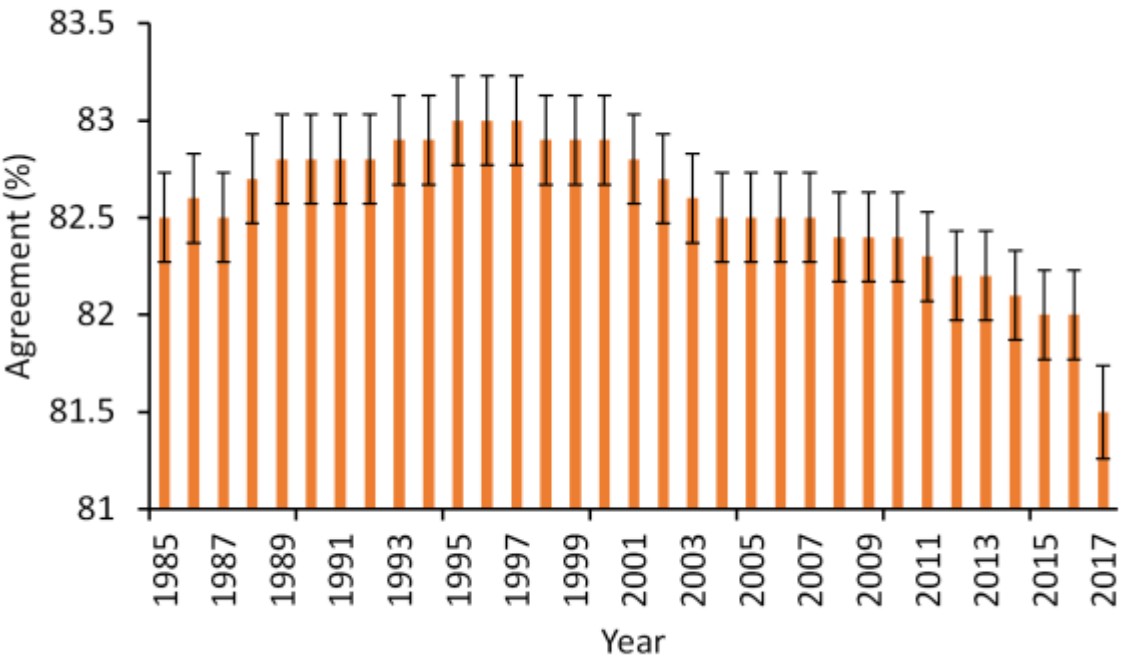

Figure 10 Overall agreement between LCMAP primary land cover and reference data across CONUS. The cross lines represent +/- one standard errors.