# Peer review of "Implementation of CCDC to produce the LCMAP Collection 1.0 annual land surface"

_Earth System Science Data, 2021_

## Author Response (AR1)

General comments:

Xian et al. present the new Landsat-based LCMAP annual 30-meter land cover and change dataset for the conterminous United States, spanning 1985-2017 (which on the LCMAP website now seems to go through 2019). Based on the state-of-the-art Continuous Change Detection and Classification (CCDC) algorithm, the product represents a large step forward in semi-operational, large-scale monitoring of land cover and its change through time. The data are well described and easily accessible, with well-defined accuracy and uncertainties (both in the manuscript itself and in the data layers and metadata). This clearly represents an important and broadly applicable resource for a wide range of applications. I have a few minor suggestions for further clarification and on the presentation in the manuscript, but overall, this is an excellent contribution that will almost certainly become a very important dataset for both science and management.

*Reply: Appreciate your comments.*

Specific comments:

Lines 103-105: Are there specific examples that the authors could provide here?

*Reply: We added several references to enhance the continuous monitoring approach. Thanks for the suggestion.*

Lines 153-155: The Introduction ends on a bit of a weak note. I'd suggest closing with a clearer statement of the objectives of the manuscript and the bigger-picture importance of the dataset, particularly following up on how the "Lessons learned" (Brown et al. 2020) informed the implementation of the LCMAP data presented here in this manuscript.

*Reply: Agree. These sentences were changed by including "Lessons learned" from prototype development that was presented in a previous publication.*

Lines 312-314 and 519-521: Why were grass and shrub included in a single grass/shrub category rather than separated into two classes? It seems like grass and shrub would likely be both ecologically and spectrally distinct, so some discussion of why that decision was made would be helpful.

*Reply: We added several sentences in 312-314 to explain the reason of combing grass and shrub together with consideration of challenges in spectral feature in the western US. Also, in the discussion section (519-521), we explained uncertainties of NLCD 2001 grass and shrub especially in the western US. The LCMAP product has achieved consistent and relatively higher accuracy by combining grass and shrub as a single class in the classification.*

Lines 472-473 and Fig. 9: Could the authors discuss or speculate about why the overall accuracy seems to decrease monotonically through time (albeit, quite a small decrease) starting around 1997? They address why the accuracy in 2017 decreases quite suddenly (limited Landsat observations at the end of the time series), but it's not apparent to me why there would be a long-term monotonic decrease prior to that rather than exhibiting more-or-less random variation from year-to-year.

*Reply: It is complicated for causes of the overall accuracy change. We believed that several factors could influence the overall agreement between reference data and mapping result. Factors including the CCDC models, Landsat data quantity and quality, training data used, the availability of high-resolution image*

*used by interpreters to generate the reference data, and specific land cover type could impact the overall validation accuracy. We added several sentences to explain potential reasons caused the overall accuracy.*

Fig. 5b, 7d-f, and 8d-f: I find it very difficult to interpret these figures. The mix of gray-scale confidence shading is tough to distinguish from the different colors, and it is nearly impossible to distinguish the two shades of purple in the maps for increasing and decreasing vegetation. Those two colors are so similar to each other that it is extremely difficult to tell them apart in the maps.

*Reply: We modified all graphics of land cover confidence by changing the dark purple color to green to make the two types of confidence more distinct.*

Fig. 5d: I would suggest rephrasing the caption to read "…(d) total number of _spectral_ changes detected…"

*Reply: Changed the words as suggested.*

Fig. 7g-h: I would suggest making the caption more descriptive about what these represent. The number of changes through time? And are these the number of thematic changes (e.g., like Fig. 5c) or spectral changes (e.g., like Fig. 5d)?

*Reply: Captions of both original Figs 7 and 8 are changed to represent the change information.*

Technical corrections:

Lines 63-64: I would suggest removing this first sentence of the Introduction. The second sentence is a much stronger opening, in my opinion.

*Reply: Moved the first sentence to the beginning of the second paragraph.*

Line 406: I don't see any dark green in Fig. 5b. (see also comment on 5b above in the specific comments)

*Reply: The graphics were changed to show green color.*

Line 416: I would suggest adding a reference to Fig. 5d after "…in the east".

*Reply: Agree. Added a reference of NLCD land cover change here.*

Line 430: I would suggest adding "(respectively)" after "2008 and 1995" to make it clearer that the increasing trend ended in 2008 for the grass/shrub and 1995 for the tree classes.

*Reply: Agee. Added the suggested word.*